# Human TIMP1 Is a Growth Factor That Improves Oocyte Developmental Competence

**DOI:** 10.3390/biotech12040060

**Published:** 2023-10-08

**Authors:** Yolanda Manríquez-Treviño, Blanca Sánchez-Ramírez, Juan Alberto Grado-Ahuir, Beatriz Castro-Valenzuela, Carmen González-Horta, M.Eduviges Burrola-Barraza

**Affiliations:** 1Facultad de Zootecnia y Ecología, Universidad Autónoma de Chihuahua (UACH), Perif. Fco. R. Almada Km. 1, Chihuahua 31453, Chihuahua, Mexico; elimanriquez.t@gmail.com (Y.M.-T.); agrado@uach.mx (J.A.G.-A.); bcastro@uach.mx (B.C.-V.); 2Facultad de Ciencias Químicas, Universidad Autónoma de Chihuahua (UACH), Campus Universitario #2, Chihuahua 31125, Chihuahua, Mexico; bsanche@uach.mx (B.S.-R.); carmengonzalez@uach.mx (C.G.-H.)

**Keywords:** TIMP1, oocyte competence, oocyte quality, oocyte maturation, in vitro maturation

## Abstract

Oocyte developmental competence is the ability of a mature oocyte to be fertilized and subsequently support embryonic development. Such competence is gained during folliculogenesis and is facilitated by the bidirectional communication into a compacted cumulus–oocyte complex (COC). Human tissue inhibitor of metalloproteinases-1 (TIMP1) participates in biological processes, including cell growth, differentiation, and apoptosis. This study aimed to evaluate the influence of TIMP1 as a growth factor on the in vitro maturation (IVM) culture of bovine COCs to improve oocyte developmental competence. All TIMP1 treatments (50, 100, and 150 ng/mL) favored the COCs’ compaction structure (*p* < 0.05). TIMP1 at 150 ng/mL produced more oocytes in metaphase II compared to the other treatments (*p* < 0.05). The 150 ng/mL TIMP1 generated oocytes with the most (*p* < 0.05) cortical granules below the plasma membrane (pattern I). In a parthenogenesis assay, oocyte IVM in 50 ng/mL of TIMP1 produced the most blastocyst compared to the other treatments (*p* < 0.05). The Principal Component Analysis (PCA) showed that 50 ng/mL of TIMP1 was the best condition to develop oocyte competence because it was associated with the COC compact and cortical granule pattern I. TIMP1 influences the development of oocyte competence when added to the IVM culture medium of COCs.

## 1. Introduction

Oocyte developmental competence refers to oocyte quality, defined as the oocyte’s ability to resume meiosis and complete its maturation. As a result, the oocyte is successfully fertilized and can support the transition to embryonic development. Competence is acquired during the growth of the oocyte in folliculogenesis, where it undergoes rearrangements at the nuclear and cytoplasmic levels [1]. During this time, the follicle cells neighboring the oocyte undergo proliferation and differentiation, forming the well-known cumulus–oocyte complex. Into the COC, the oocyte and the surrounding cumulus cells maintain a tight connection through bidirectional communication that leads to reached oocyte maturity. Once the oocyte has matured, the COC leaves the follicle and enters the oviduct to be fertilized [2]. For COC to fulfill this task, the cumulus cells must expand through a specialized extracellular matrix (ECM) assembly between the cells that make up the COC. The ECM conformation begins after the luteinizing hormone (LH) surge and by the action of oocyte-secreted factors like GFP9 and BMP15. Therefore, ECM establishment and preservation into the COC is vital to successful ovulation and fertilization [3].

TIMP1 is a member of the tissue inhibitor of the metalloproteinases (TIMPs) family of proteins. This family inhibits the action of matrix metalloproteinases (MMPs) and disintegrin-metalloproteinases as ADAM-10, leading to the reorganization of the ECM. The classical role of TIMPs suggests that an increase in these proteins results in ECM accumulation, and conversely, a decrease in TIMPs leads to an increase in ECM proteolysis [4]. In addition to its inhibitory effect on MMPs, TIMP1 participates in multiple processes such as cell proliferation, cell survival and differentiation, and apoptosis [5,6]. Over the last three decades, studies have demonstrated that TIMP1 is a ubiquitous protein expressed in widespread mammalian tissues, remarkably in the reproductive organs [7,8,9,10,11,12,13,14]. In goats, TIMP1 was detected in the oocytes, cumulus, granulosa, and theca cells of antral follicles and corpus luteum [14]. Particularly in the porcine and bovine model, the presence of exogenous TIMP1 in the culture media of COCs during in vitro maturation (IVM) increases the capacity of oocytes to develop blastocysts [10,11]. Also, TIMP1 augments estradiol secretion in swine theca and goat granulosa cell cultures [12,14]. Finally, through 24 h of an IVM culture of bovine COCs, TIMP1 expression increases in the last 12 h of the culture [15].

Since TIMP1 acts as an MMP inhibitor with the consequence of a positive effect on maintaining ECM—which in the COC microenvironment, implies that cumulus cells expand adequately—it is supposed that a high amount of TIMP1 could improve this phenomenon, favoring the competence of oocyte development. Thus, the objective of this study was to evaluate the influence of TIMP1 as a growth factor on the in vitro maturation (IVM) culture media of bovine COCs to improve oocyte developmental competence.

## 2. Materials and Methods

### 2.1. Reagents and Media

The reagents used for the culture medium were from Sigma-Aldrich, St. Louis, MO, USA; the catalog number of each reagent is within parentheses unless otherwise indicated.

### 2.2. Bioinformatic Analysis of TIMP-1 Amino Acid Sequences

The TIMP1 amino acid sequences for both the human and bovine proteins were obtained from the Protein Bank at the National Center for Biotechnology Information, NCBI, https://www.ncbi.nlm.nih.gov/ (accessed on 15 November 2017), with accession numbers NP_003245.1 and NP_776896.1, respectively. The Blast program of NCBI was used to find the percent identity between both sequences, and the Clustal Omega program of The European Bioinformatics Institute, EMBL-EBI, http://www.ebi.ac.uk/Tools/msa/clustalo/ (accessed on 15 November 2017) was used for the alignment. The amino acid tertiary structure was obtained using the Robetta protein predictor service, http://robetta.bakerlab.org (accessed on 17 November 2017). The phylogenetic analysis was performed using the function “build” of ETE3 3.1.2 [16] as implemented on the GenomeNet, https://www.genome.jp/tools/ete/ (accessed on 20 November 2017). The tree was constructed using fasttree with slow NNI and MLACC = 3 (to make the maximum-likelihood NNIs more exhaustive) [17].

### 2.3. COCs In Vitro Maturation (IVM)

Bovine ovaries were obtained from a local slaughterhouse and transported to the laboratory within 2 h in a 0.9% sterile saline solution. The COCs were aspirated from follicles ranging from 2.0 to 6.0 mm in diameter using a PrecisionGlide needle (BD 18G × 1½”) by applying constant pressure (~50 mmHg) with a vacuum pump (WOB-L^®^ Dry Vacuum Pumps, Standard-Duty, Welch^®^). The aspiration needle was connected to a sterile 50 mL plastic tube. The COCs were washed twice in a chemically defined medium, CDM [71 mM NaCl (S5886), 6 mM KCl (P5405), 1 mM KH_2_PO_4_ (P5655), 0.5 mM Na-citrate (C85532), 25 mM NaHCO_3_ (S5761), 2 mM CaCl_2_ (C5670), 4.9 mM glycine (G8790), 1 mM alanyl-glutamine (A8185), 20 mM HEPES (H4034), 10 mM sodium L-lactate (L7022), 0.5 mM Na-pyruvate (P5280), 0.5 mM MgSO_4_ (M2643), 0.67 mM non-essential amino acids (M7145), and 25 μg/mL gentamycin (G1264)] and transferred to handling medium, H-CDM m [CDM supplemented with 0.5 mM D-fructose (F3510), 2.5% fatty acid-free BSA (A6003), 22.5 mM NaCl (S5886), and 20 μg/mL heparin Na salt (H3149)] for the manipulation of oocytes. COCs selected for study with at least three layers of cumulus cells and uniform cytoplasm were visualized using a stereomicroscope (Leica MS5, Leica Microsystems GmbH, Wetzlar, Germany). COCs were cultured in four-well plates (Nunc, Thermo Scientific, Rockford, IL, USA). Briefly, groups of approximately 50 COCs per well were maintained in 1 mL of chemically defined medium for maturation, M-CDM [CDM supplemented with 2 mM D-fructose (F3510), 2.77 mM myo-inositol (I7508), 0.1 mM taurine (T8691), 5% fatty acid-free BSA (A6003), 15 ng/mL of FSH (F4021), 0.1 μg/μL 17β-estradiol (E2257), 50 ng/μL of EGF (E9644), and 0.1 mM cysteamine (M6500)]. The COCs were cultured for 24 h in 5% CO_2_ in humid air at 38.5 °C. For all experiments, the control medium used was M-CDM, and the test medium was M-CDM plus TIMP1 human recombinant protein (SRP6040) at 50, 100, or 150 ng/mL. The concentration of TIMP1 doses was chosen according to Shores & Hunter (2000).

### 2.4. Appraisal of COC Compaction

The proportions of COCs exhibiting different degrees of cumulus compaction were determined at the end of IVM. The degree of cumulus compaction was assessed with the cumulus cells’ morphology as visualized using an Axiovert CFL-40 inverted microscope (Carl Zeiss, Inc., Göttingen, Germany), according to Kobayashi et al. [18]: compacted if cumulus cells were very clustered without any separation between them; partially compacted if cumulus cells were not homogeneously spread and clustered cells were still observed; and expanded when cumulus cells were homogeneously spread and clustered cells were no longer present.

### 2.5. Evaluation of Nuclear Maturation

The oocytes were collected from COCs after 24 h of IVM and were denuded by vortexing with 100 μl hyaluronidase (1000 U/mL, cat. no. H3631) in phosphate-buffered saline (PBS) for 5 min, then washed in 1% polyvinyl alcohol (PVA). Next, the oocytes were permeabilized with 0.2% Triton X-100 (T8787) in PBS-PVA, then stained with propidium iodide (15 μg/mL, cat. no. P4170) for 5 min at 37 °C. Finally, the oocytes were placed on a microscope slide in a 10-μL drop of glycerol and observed using an Axio Imager.M2 fluorescence microscope (Carl Zeiss, Inc., Göttingen, Germany) at 617 nm for nuclear staining. Images were acquired using AxioVision Software and an AxioCam MRm Digital Camera (Carl Zeiss, Inc., Göttingen, Germany). Each oocyte was evaluated for the stage of nuclear maturation as follows: germinal vesicle (GV), germinal vesicle breakdown (GVBD), or metaphase II (MII).

### 2.6. Cortical Granule Staining

The method for staining cortical granules was based on those reported by Yoshida et al. [19], with a few modifications. First, in vitro-matured oocytes were denuded of cumulus cells via vortexing with 0.5% pronase (P8811), washed three times in PBS containing 0.4% BSA, and then fixed with 3.7% paraformaldehyde (P6148) for 30 min at room temperature. The oocytes were blocked by washing them three times in a solution of 0.3% BSA and 100 mM glycine (G8790). Then, they were permeabilized with 0.1% Triton X-100 (T8787) for 5 min and incubated in 20 μg/mL fluorescein isothiocyanate (FITC)-labeled lens culinaris agglutinin (LCA) (Invitrogen, Grand Island, NY, USA). Finally, the oocytes were mounted on a slide in a 10-μL drop of glycerol and observed using an Axio Imager.M2 fluorescence microscope (Carl Zeiss, Inc., Göttingen, Germany) at 515 nm for fluorescent CGs. Images were acquired using AxioVision Software and an AxioCam MRm digital camera (Carl Zeiss, Inc., Göttingen, Germany).

### 2.7. Parthenogenetic Activation and Embryo Culture

Matured oocytes were denuded via vortexing with 100 μl hyaluronidase (1000 U/mL) and then activated by incubation in 0.2 mM calcium ionophore (C7522) for 5 min, followed by treatment with 2 mM 6-DMAP (S2629). Then, parthenogenetically activated presumed zygotes were cultured in 400 μL of a chemically defined medium for the in vitro culture of embryos CDM-2 [CDM supplemented with 2 mM D-fructose (F0127), 2.77 mM myo-inositol (I5125), 1.47 mM essential amino acids (M5550), 5% BSA (A9576) and 5 mM NaCl (S5886)] in 4-well plates (Nunc, Thermo Scientific, Rockford, IL, USA). One mL of purified water was added to the center hole of the plate, and the embryos were then incubated with 5% CO_2_, 5% O_2_, and 90% N_2_ in humidity-saturated air at 38.5 °C for seven days.

### 2.8. Statistical Analysis

Statistical analyses were performed using XLSTAT Software [20] and Jamovi Software [21]. The data for cumulus cell compaction, oocyte maturation, cortical granule distribution, and embryos developed were expressed as percentages. These data were subjected to one-way ANOVA and a means comparison was obtained using Tukey’s test. The significance level was set at *p* < 0.05, while 0.05 ≤ *p* ≤ 0.10 was considered a trend. All values are presented as means with their corresponding standard error. A Principal Component Analysis (PCA) was applied using a varimax rotation to reduce the dimensionality and detect essential patterns in data. Before analysis, the data were standardized, resting the media, and divided by the standard deviation. After a rotation with varimax, sample adequacy was questioned by the Kaiser–Meyer–Olkin (KMO) test (<0.5), and Bartlett’s test of Sphericity evaluated the suitability for component analysis. The components were identified using an eigenvalue threshold of >0.1. The results were shown in a biplot graphic.

## 3. Results

### 3.1. Comparison of Bovine and Human TIMP1 Protein

Amino acid sequence analysis reveals 87% homology between bovine and human TIMP1. Furthermore, both proteins have the same tertiary structural conformation (Figure 1), which corresponds to the structure reported by Grünwald et al. [5], showing an N-terminal domain (AA 1–126) and a C-terminal domain (AA 127–184), where the N-terminal domain is required for the inhibition of MMPs. The phylogenetic tree demonstrated that human and bovine TIMP1 proteins were closely positioned in the same node, and both were very distanced from the mouse TIMP1 localized in the other branch.

### 3.2. Effect of TIMP1 on the Degree of COC Compaction during IVM

Figure 2A shows a representative photomicrograph of COCs after 24 h of IVM to evaluate the degree of compaction. A difference from culture control, where there was no statistical significance between compaction structures (*p* > 0.05); in all TIMP1 treatments, the compacted COCs were the most abundant structure. In the 50 ng/mL TIMP1 condition, the percentage of compacted and partially compacted COCs were close, while the expanded COC was the lower structure (57 ± 9.8% and 38 ± 8.6% vs. 4.5 ± 1.9%, respectively; *p* < 0.05). A condition with 100 ng/mL of TIMP1 showed that the higher structure was COC compacted compared with the COCs’ partially compacted and expanded assemblies (64 ± 5% vs. 27 ± 6% and 9 ± 2%, respectively; *p* < 0.05). The same was for the 150 ng/mL TIMP1 group; the COC structure was more abundant than the others (50 ± 9% vs. 34 ± 8% and 16 ± 8%, respectively; *p* < 0.05).

### 3.3. Effect of TIMP1 on Oocyte Nuclear Maturation during IVM

Figure 3A shows representative images of propidium iodide-stained oocytes for nuclear observation to classify the stage of oocyte maturation (GV, GVBD, and metaphase II) at the end of IVM. As shown in Figure 3B, 150 ng/mL of TIMP1 produced the highest proportion of oocytes arrested at MII compared with the other treatments (60 ± 5.3 for 150 ng/mL, 45 ± 1.5 for 50 ng/mL, and 29 ± 0.7 for 100 ng/mL and 21 ± 1.5 for the control; *p* < 0.05). This treatment also produced the lowest proportion of oocytes at the GV stage (17 ± 2.5 vs. 45 ± 2.9 in the control group, 35 ± 2 in the 50 ng/mL group, and 52 ± 2% in the 100 ng/mL group; *p* < 0.05). Finally, the proportion of oocytes at the GVBD stage was the highest in the control group (33 ± 2% vs. approximately 20% for each of the three TIMP1 treatments; *p* < 0.05).

### 3.4. Effect of TIMP1 on Cortical Granules Distribution

The cortical granule distribution patterns in oocytes after IVM were classified into two patterns: (I) oocyte mature if the cortical granules were distributed adjacent to the plasma membrane as a monolayer; (II) oocyte immature if the cortical granules were distributed in the cortical area away from the plasma membrane and formed clusters in the cytoplasm (Figure 4A). The 150 ng/mL TIMP1 treatment was the only treatment that produced a significant difference in the distribution of cortical granules (Figure 4B); in this case, the oocytes showed a higher proportion of pattern I than pattern II (55 ± 0.6% vs. 44 ± 0.33%, respectively, *p* < 0.05).

### 3.5. Early Embryo Development

Early embryo development was induced via oocyte parthenogenetic activation. Figure 5 shows each treatment’s effect on the embryo development rates. Regarding cleavage, all treatments and the control showed similar proportions of 2–4 cell embryos (36 ± 1.8%, 45 ± 2.5%, 38 ± 5.9%, 41 ± 3.2%, for 50, 100, 150 ng/mL TIMP1 and control, respectively; *p* > 0.05). On the other hand, the 150 ng/mL treatment generated the highest proportion of 8- to 16-cell embryos compared with the other treatments (21 ± 1.9%, 22 ± 3.2%, 30 ± 0.3%, 24 ± 0.7%, for 50, 100, 150 ng/mL TIMP1 and control, respectively *p* < 0.05). Instead, the 50 ng/mL treatment produced the most blastocyst compared to the other treatments (43 ± 2.2%, 32 ± 4.3%, 32 ± 3.0%, 35 ± 3.8%, for 50, 100, 150 ng/mL TIMP1 and control, respectively *p* < 0.05).

### 3.6. Relationship between Oocyte Quality and Treatment Variants

The Bartlett test of Sphericity was significant (X^2^ = 35.6, df = 5, *p* < 0.001), which implies that the observations of COC compaction structures, oocyte nuclear maturation, cortical pattern type, and parthenogenetic embryos were related to treatment variables and that PCA could be used to analyze multiple associations between them. The KMO test showed values between 0.7 and 0.8, indicating sample adequacy. The biplot graph displayed two principal components (PCs); PC1 absorbed 87% of the variables, while PC2 absorbed 7.4%; both add up to 94.4%; this implies sufficient information existed to interpret the data’s most important aspects (Figure 6). In the biplot graph, the observations of expanded COCs, the 8–16 cells, 2–4 cells, and blastocyst embryos were located closer to each other with lower PC scores and were not associated with any treatment. Oocyte GV and oocyte GVBD had high PC scores but were not associated with any treatment. The observations for partially compacted COCs and cortical granule pattern II were located together and were associated with the control culture (0 ng/mL) and 100 ng/mL TIMP1 treatment. For 150 ng/mL of TIMP1, this variable was projected near cortical granule pattern I. Likewise, the cortical granule pattern I and compacted COC were located in the same place and were related to the 50 ng/mL TIMP1 treatment.

## 4. Discussion

Oocyte developmental competence is the ability of a mature oocyte to be fertilized and subsequently lead to the early development of the embryo. Such competence is acquired during oocyte growth in a follicle, where the oocyte and its surrounding cells maintain a strongly bidirectional communication [1]. As a result, a COC is conformed in the follicular microenvironment at the appropriate time, and the oocyte culminates its maturation. An oocyte is mature when it can arrest its meiosis in metaphase II (MII) after having reorganized the organelles in its cytoplasm and stored mRNA, proteins, and transcription factors that must be required to support the first stages of embryonic development [22,23]. In addition, oocyte maturation involves structural changes that lead to follicle rupture with subsequent expulsion of COC from the follicle [24]. The release of a mature oocyte from the follicle is a complicated process that implies the conformation in the COC of an ECM network consisting of cross-linked hyaluronan, PTX3, TNFAIP6, and IαI proteins. The ECM caused the expansion of cumulus cells, which provoked the loss of bidirectional communication within the COC [25]. With an expanded structure, the COC will be released and reach the oviduct, where the oocyte is fertilized [26]. Thus, the maintenance of the ECM is vital for these processes. Here, TIMP1 contributes since its inhibitory action on MMPs ensures that the extracellular network remains intact.

In vivo, after hormonal stimulation by LH and before the rupture of the follicular wall, the granulosa cells begin to synthesize TIMP1, which is secreted and accumulates in the follicular fluid of antral follicles [27]. At the moment of rupture, the elevated amount of TIMP1 in the follicular fluid deactivates the action of MMPs in the follicular wall apex region, leading to the differentiation of granulosa cells in luteal cells [28]. In vitro, the *timp1* expression has an actively high expression after 12 h of IVM in bovine COCs, opposing the *mmp1* expression, significantly decreasing after that culture hour [15]. This fine-tuned balance between MMPs and TIMP1 suggests that TIMP1 maintains the network of ECM between the COCs, which leads to cumulus cell expansion and, subsequently, the development of oocyte competence. With this in mind, the route of this work was to confirm that TIMP1 promotes the structure expanded into the COC. In this work, we used human TIMP1 protein instead of bovine TIMP1, because it was available in the laboratory. TIMP1 is a highly conserved 28 kDa protein in mammals and is composed of two domains that are stabilized by three disulfide bonds: an N-terminal domain and a C-terminal domain. The N-terminal domain of approximately 125 amino acids possesses inhibitor activity against MMPs, ADAM-10, and ADAMTS4 proteases and binds to the CD82 receptor. The C-terminal domain of about 65 amino acids strongly interacts with the CD63 receptor and proMMP-9 [5,29]. Our in silico comparison of bovine and human TIMP1 proteins revealed they share 87% amino acid sequence homology and have virtually identical three-dimensional conformations. Nishimura et al. [30] showed that human or bovine TIPM1 proteins were able to inhibit the metalloproteinase (MMP) activities in the Gin-1 (human) or 3T3 (mouse) cells. However, interestingly, human TIMP1 could not stimulate the growth in mouse cells, and this effect was just for human cells. Conversely, bovine TIMP1 stimulated the proliferation in both human and mouse cells. Nevertheless, bovine TIMP1 could provoke human cell proliferation, which means that in human cells, bovine TIMP1 is recognized by the same receptors that recognize human TIMP1. Hence, it is probable that this is the same for human TIMP1 in bovine cells. With this in mind, we executed a phylogenetic analysis. The results showed that human and mouse TIMP1 had a 74% amino acid sequence homology, lower than human and bovine TIMP1 homology. Also, the phylogenetic tree demonstrated that human and bovine TIMP1 proteins were closely positioned in the same node, and both were significantly distanced from the mouse TIMP1 localized in the other branch. With the assumption that human TIMP1 has the same action as bovine TIMP1, for all the experiments of this work, we used readily available recombinant human TIMP1.

The most popular function of TIMP1 protein is that it acts as an MMP inhibitor, which maintains the ECM necessary for COC expansion. Our results showed that COCs’ expanded structures were not favored by TIMP1 action. On the contrary, TIMP1 favored compacted COC assemblies. A compact COC structure favors bidirectional communication between the oocyte and the cumulus cells surrounding it, which is critical for the oocyte to acquire the necessary competence to support oocyte-to-embryo transition [31]. Several studies have shown that, independently of its inhibitory action on MMPs, TIMP1 regulates cell proliferation, survival, and differentiation [5,6]. TIMP1 acts as a promoter of cell proliferation in a wide array of cell types, including granulosa cells, fibroblasts, epithelial cells, carcinoma cells, keratinocytes, and leukemic cell lines [14,32]. Interestingly, Hong et al. [14] showed that in goat granulosa cells, the overexpression of the *timp1* gene, besides promoting cellular proliferation, also encouraged estradiol secretion by these cells. Translating this finding to our results, it is likely that TIMP1 promotes the proliferation of cumulus cells, which could support the COC compaction structure. How can this be explained?

On the one hand, TIMP1 controls the activity of MMPs; on the other, it stimulates cellular proliferation. TIMP1 has two independent domains, and these reside in its multi-functionality. First, TIMP1 acts with its N-terminal domain as an inhibitor of MMP, and ECM conformation occurs. Second, if there is a high level of TIMP1, this favors an increased concentration of free uncomplexed TIMP1, which, using its C-terminal, can bind to CD63 and CD82 in the membrane of a target cell. CD63 and CD82 are membrane receptors activating signals via MAPK, ERK, ILK, FAK, integrins, Src, and c-Kit, which regulate cell proliferation and growth [33,34]. Another explanation resides in the inhibitory action of TIMP1 over ADAM-10 functions [4]. ADAM-10 is a membrane-spanning protease composed of one transmembrane domain that connects a short cytoplasmic C-terminal tail with an extracellular domain. ADAM-10 is responsible for shedding membrane-spanning proteins by cutting off the external domain and releasing it as active ectodomains, which can act in target cells, leading to a downregulating signaling pathway [35]. In COCs in bovines, the *ADAM-10* gene is expressed by the action of BMP15 and FGF10 oocyte-secreted factors [36]. Following the LH surge, it binds to its receptor in mural granulosa cells of the preovulatory follicle. Then, it activates all the signal pathways needed to synthesize amphiregulin, epiregulin, and betacellulin proteins. Once these proteins are produced, they are driven to be inserted into the membrane as membrane-anchored precursors; then ADAM-10 cuts off their extracellular domain, releasing their soluble ectodomain into the intrafollicular space. Subsequently, they are anchored to EGFR in the cumulus cell membrane, triggering the synthesis of the PTGS2, HAS2, PTX3, and TNFA1P6 proteins required for ECM remodeling during cumulus cell expansion [37]. Accordingly, we propose that a high level of TIMP1 inhibited the ADAM-10 shedding action in cumulus cells during the IVM. Consequently, ECM assembly does not occur, the expansion of cumulus cells is blocked, and the COC maintains its compacted structure.

Nuclear maturation occurs as a consequence of stimulation by LH. The first morphological sign of resumption of meiosis is the GV rupture to form GVBD. Post-GVBD, the meiosis I spindle is formed, and bivalent chromosomes are aligned on the metaphase plate, leading to anaphase I. Meiosis I is completed when the oocyte extrudes a polar body containing a set of chromosomes. Subsequently, the oocyte enters a second round of meiosis and is arrested at MII until fertilized. Our results show that during IVM, the addition of TIMP1 at a dose of 150 ng/mL favors the development of oocytes in MII in a way that increases nuclear maturation threefold compared with the control. This result agrees with Funahashi et al. [11], who showed that adding TIMP1 to the IVM culture medium favored the development of mature oocytes in pigs. Similarly, Ayvazova et al. [38] demonstrated that human follicles with higher concentrations of TIMP1 generated more oocytes in MII than those with low TIMP1.

In cytoplasmic maturation, oocyte organelles such as mitochondria, endoplasmic reticulum, cortical granules, and the Golgi apparatus are distributed at different locations in the cytoplasm during the transition from GV to MII. A poor distribution of organelles implies poor cytoplasmic quality, which harms competence development [1]. Cortical granules are membranous vesicles derived from the Golgi apparatus, exclusive of the oocyte, whose composition includes proteins, structural molecules, enzymes, and glycosaminoglycans. In GV-stage bovine oocytes, the cortical granules are distributed in clusters throughout the cytoplasm [39,40]. In contrast, in oocytes that have reached MII, the granules are distributed as a monolayer on the inner face of the plasma membrane, waiting to be released when the plasma membrane of a sperm fuses with the oocyte membrane [41]. When that occurs, intracellular signaling is unleashed within the oocyte to stimulate the exocytosis of the cortical granules, which then release their contents to block the union of more sperm [40]. Our results show that the supplementation of IVM medium with 150 ng/mL of TIMP1 causes the cortical granules to be adequately distributed at the end of maturation, forming a monolayer below the plasma membrane—an indication of correct cytoplasmic maturation in the oocyte.

After fertilization, the onset of embryonic development depends on the quality that the oocyte attained during competence development. The parthenogenetic is a process where an oocyte, without sperm fertilization, can develop an embryo. Therefore, a parthenogenetic assay is an excellent way to evaluate oocyte competence because there is no influence of sperm. With a parthenogenetic assay, all the cellular changes in the oocyte-to-embryo transition can be exclusively awarded to the oocyte. Our results showed that the dose of 50 ng/mL of TIMP1 influenced the competence of oocytes after their maturation, which was directly reflected in the higher parthenogenetic blastocyst development. Hence, TIMP1 favors the development of oocyte competence. In turn, these oocytes can be activated and stimulated to develop by parthenogenesis up to the blastocyst stage, indicating they are competent to transition toward the development of an embryo.

To clarify and obtain an overview of how the observations of COC compaction structures, oocyte nuclear maturation, cortical pattern type, and parthenogenetic embryos were related to TIMP1 treatment variables, a PCA was used to analyze multiple associations between them. After that, the result inferred that the treatment of 50 ng/mL was the best condition to develop oocyte competence, because it promotes the formation of COC compaction structure and the correct location of the cortical granules. Accordingly, we recommended supplementing the culture IVM media with 50 ng/mL of TIMP1 human recombinant protein.

It is essential to mention that this study had two limitations. First, there were only three treatments of TIMP1: 50, 100, and 150 ng/mL. TIMP1 protein functions as an MMP inhibitor, supporting the ECM necessary for COC expansion. In this study, the COCs’ expanded structures were not favored by TIMP1 action. In contrast, the presence of TIMP1 promoted compacted COCs. Nonetheless, Figure 2B shows that the percentage of expanded COCs slightly increased in a TIMP1 dose-dependent (4.5 ± 5.9, 9.3 ± 4.6, and 16.3 ± 8.7 for 50, 100, and 150 ng/mL of TIMP1, respectively). This result may suggest that the TIMP1 concentration was insufficient to conduce to the COCs’ expansion. Hence, another studio is necessary using TIMP1 concentration higher than 150 ng/mL. Second, early embryo development was not measured after an in vitro fertilization (IVF) with a sperm. Although the parthenogenetic assay is an excellent way to evaluate the oocyte quality, an IVF is needed to evaluate the influence of TIMP1 over the yield of blastocysts. Therefore, further analysis is needed to solve these limitations.

## 5. Conclusions

TIMP1 is a protein that, when added in a concentration of 50 ng/mL to the IVM culture medium of COCs, positively influences the development of oocyte competence. The presence of TIMP1 favors the compaction of COCs. It also improves nuclear maturation since it enhances the formation of MII-stage oocytes. TIMP1 also promotes correct oocyte cytoplasmic maturation, because it increases the distribution of cortical granules below the plasma membrane. In this way, oocytes IVM in the presence of TIMP1 can be activated and readied to oocyte-to-embryo transition, as reflected by the increased production of parthenogenetic blastocysts.

## Figures and Tables

**Figure 1 biotech-12-00060-f001:**
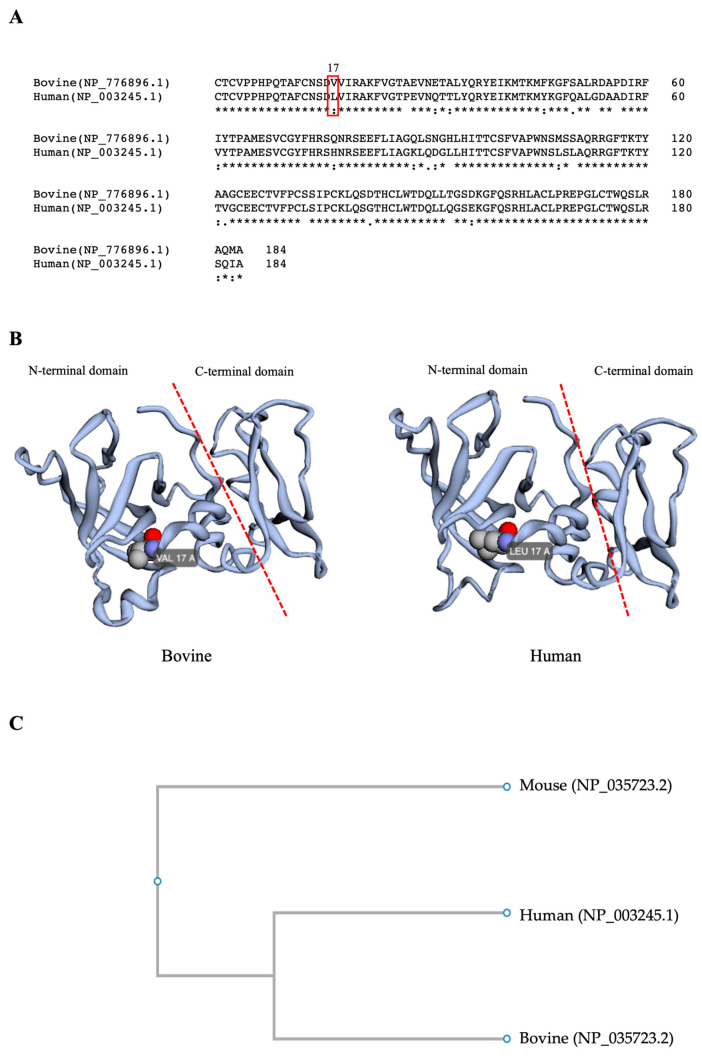
A comparison of bovine and human TIMP1 proteins. (**A**). The protein sequence alignment of bovine and human TIMP-1; accession numbers are in parentheses. The red rectangle indicates the amino acid at position 17 from the N-terminus. (**B**) Bovine and human TIMP1 proteins’ tertiary structures show the amino acid at position 17; VAL = valine, LEU = leucine. (**C**) A phylogenetic analysis of mouse, human, and bovine TIMP1 proteins. The access number to the Protein Bank is inside the parenthesis.

**Figure 2 biotech-12-00060-f002:**
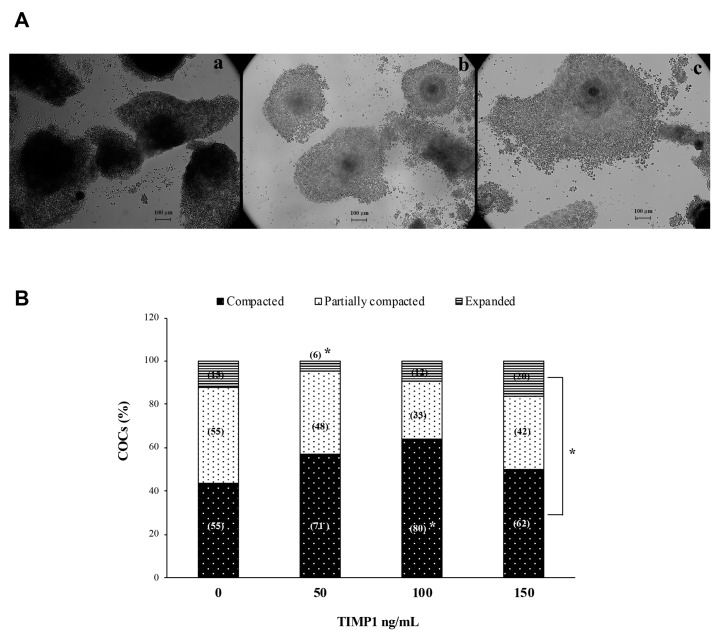
The effect of different concentrations of TIMP1 on the degree of cumulus cell compaction. (**A**) (**a**): Compacted; (**b**): Partially compacted; (**c**): Expanded. The compaction degree of the COCs was evaluated after 24 h of IVM. Images were visualized at 10× with an inverted microscope Axiovert CFL40 (Zeiss). (**B**) Comparison between the degree of cumulus cell compaction at different concentrations of TIMP1. Data are represented as the mean ± standard error percentage from 500 COCs, where each treatment had 125 distributed in 5 independent experiments with 25 COCs each. The number in the parenthesis in each bar is the total of COCs for each variant. The asterisk, *, represents a statistically significant difference (*p* < 0.05).

**Figure 3 biotech-12-00060-f003:**
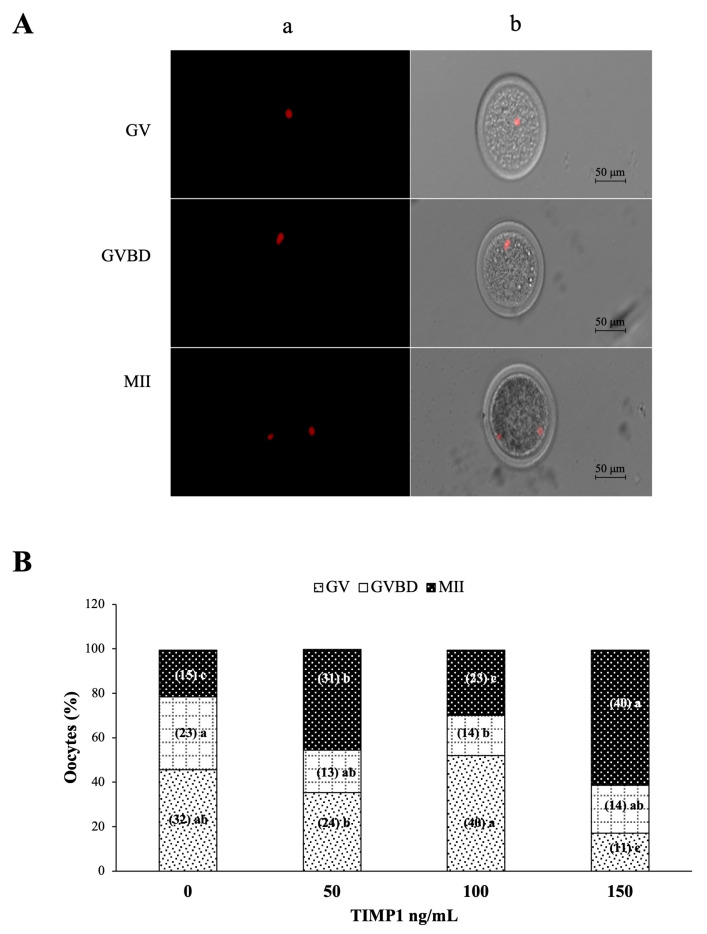
The effect of different concentrations of TIMP1 on oocyte nuclear maturation after IVM. (**A**) Representative images of the stages of nuclear maturation in the oocyte. DNA was visualized with red fluorescence after staining with propidium iodide; for each one, (**a**) propidium iodine staining only and (**b**) propidium iodine and phase-contrast merged. (**B**) Comparison between the stages of maturation with different concentrations of TIMP1. GV: germinal vesicle; GVBD: germinal vesicle breakdown; MII: metaphase II. Data are represented as the mean ± standard error percentage from 280 COCs, where each treatment had ~70 distributed in three independent experiments with ~23 COCs each. The number in the parenthesis of each bar is the total of COCs for each variant, and different letters represent a statistically significant difference (*p* < 0.05).

**Figure 4 biotech-12-00060-f004:**
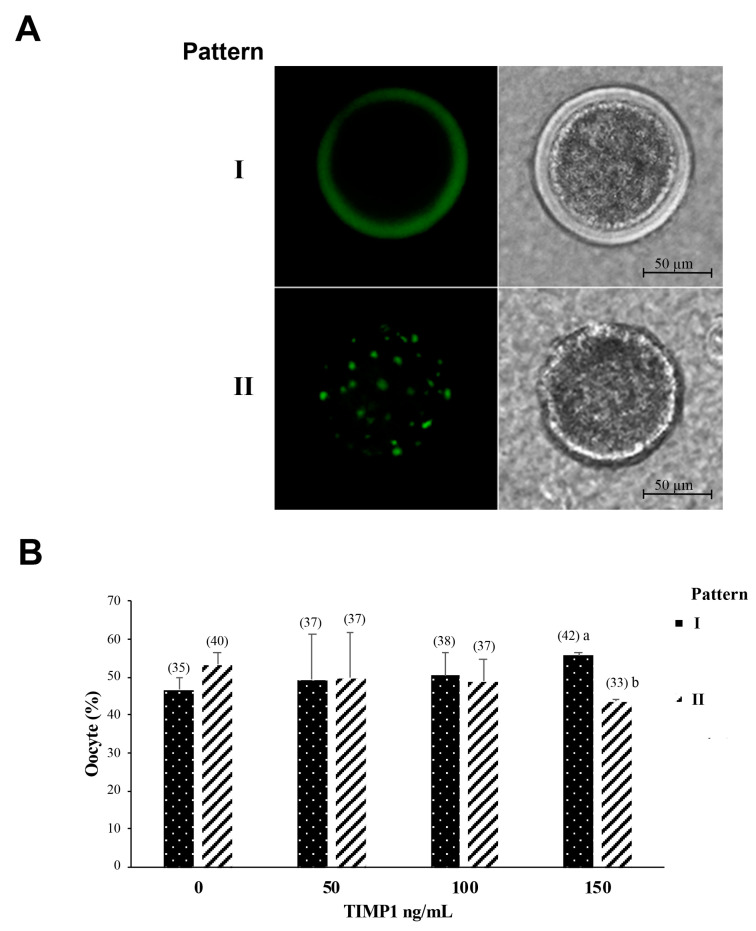
The effect of different concentrations of TIMP1 on cortical granule distribution. (**A**) A representative image of cortical granule distribution in the oocyte. The cortical granules were visualized with green fluorescence after staining with LCA-FITC. Patterns: I. cortical granules distributed adjacent to the plasma membrane, and II. Cortical granules are distributed in the cortical area. (**B**). Comparison of cortical granules distribution among different concentrations of TIMP1. Data are represented as the mean ± standard error percentage from 298 COCs, where each treatment had ~75 distributed in 3 independent experiments with ~25 COCs each. The number in the parenthesis over each bar is the total COCs for each variant, and different letters represent a statistically significant difference (*p* < 0.05).

**Figure 5 biotech-12-00060-f005:**
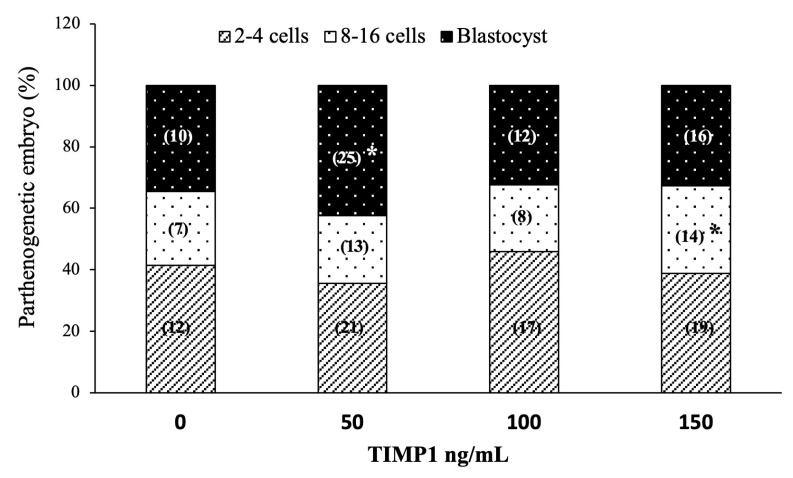
The effect of different TIMP1 concentrations during oocyte IVM on parthenogenetic embryo development. Data are represented as the mean ± standard error percentage from 174 COCs, where each treatment had ~44 distributed in three independent experiments with ~15 COCs each. The number in the parenthesis in each bar is the total of COCs for each variant. The asterisk, *, represents a statistically significant difference (*p* < 0.05).

**Figure 6 biotech-12-00060-f006:**
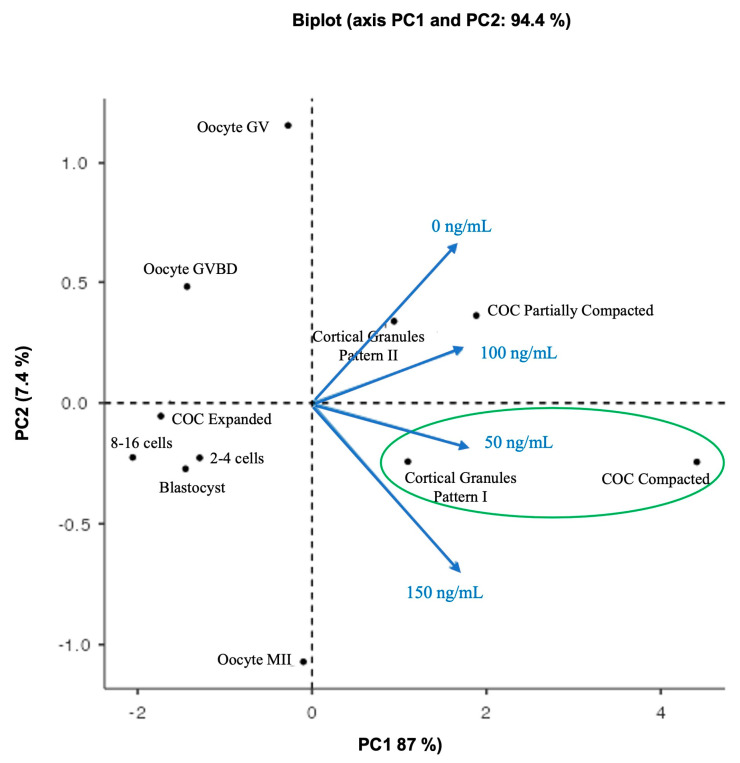
Principal component analysis (PCA) analysis biplot. The black circle represents the observations of COC compaction structures, oocyte nuclear maturation, cortical pattern type, and parthenogenetic embryos. The TIMP1’s treatment variables are presented as blue arrows that radiate from the center and reflect the directionality and strength (through the length) contribution of each.

## Data Availability

All data underlying the results are included as part of the published article. Further information can be provided by corresponding author upon request.

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
