# Peer review of "Human TIMP1 Is a Growth Factor That Improves Oocyte Developmental Competence"

_biotech, 2023, doi:10.3390/biotech12040060_

Round 1

Reviewer 1 Report

In this manuscript, the author evaluated the influence of TIMP1 on the IVM culture of bovine COCs. The data shows that TIMP1 supplementation enhanced the expanded COCs structures compared with the control group and produced the most blastocyst in parthenogenesis as well. Here are some concerns needed to be clarified.

Majors:

1. For figure 2B, 3B, 4B and 5B, I recommend the author to add the exact counting numbers for each group (column) instead of showing the percentage only, which would provide more information about the statistics.

2. For the TIMP1 concentration issue, why did the author choose 150ng/mL as the max concentration? Did the author try higher one to see if it would lead to better oocyte developmental competence?

3.  Why did the author choose to use parthenogenetic activation to evaluate the early embryo development instead of in vitro fertilization?  

4. The author should give a deeper discussion about the limitation for this research.

Minors:

1. Line 63, the ‘in vitro’ should be shown as italics, the author should check this for the whole manuscript.

2. For figure 2A ,3A and 4A, the scale bar should be added.

Author Response

RESPONSE TO REVIEWER 1 COMMENTS

Majors:

1.For Figures 2B, 3B, 4B and 5B, I recommend the author to add the exact counting numbers for each group (column) instead of showing the percentage only, which would provide more information about the statistics.

We appreciate this recommendation. The exact counting numbers of each variable were added inside a parenthesis on each figure.

2. For the TIMP1 concentration issue, why did the author choose 150ng/mL as the max concentration?

Based on the results of Funahashi et al., 1999  and Satoh et al., 1994, who evaluated TIMP1 over the oocyte competence in the porcine model, we tried to use similar TIMP1 concentrations. We had an economic limitation to buy reagents. So, we evaluated the total amount of TIMP1 that we possessed and divided it into three treatments: 50, 100, and 150 ng/mL.

References:

Funahashi, H., McIntush, E. W., Smith, M. F., & Day, B. N. (1999). The Presence of Tissue Inhibitor of Matrix Metalloproteinase-1 (TIMP-1) During Meiosis Improves Porcine “Oocyte Competence” as Determined by Early Embryonic Development After In-vitro Fertilization. Journal of Reproduction and Development, 45(4), 265–271. https://doi.org/10.1262/JRD.45.265

Satoh, T., Kobayashi, K., Yamashita, S., Kikuchi, M., Sendai, Y., & Hoshi, H. (1994). Tissue inhibitor of metalloproteinases (TIMP-1) produced by granulosa and oviduct cells enhances in vitro development of bovine embryo. Biology of Reproduction, 50(4), 835–844. https://doi.org/10.1095/biolreprod50.4.835

Did the author try higher one to see if it would lead to better oocyte developmental competence?

No. Unfortunately, we did not try a higher 150 ng/mL dose.

3. Why did the author choose to use parthenogenetic activation to evaluate the early embryo development instead of in vitro fertilization?

After fertilization, the onset of embryonic development depends on the quality that the oocyte attained during competence development. The parthenogenetic is a process where an oocyte, without sperm fertilization, can develop an embryo. Therefore, a parthenogenetic assay is an excellent way to evaluate oocyte competence because there is no influence of sperm. With a parthenogenetic assay, all the cellular changes in the oocyte-to-embryo transition can be awarded exclusively to the oocyte. Our results showed that the dose of 50 ng/mL of TIMP1 influenced the competence of oocytes after their maturation, which was directly reflected in the higher parthenogenetic blastocyst development. Hence, TIMP1 favors the development of oocyte competence. In turn, these oocytes can be activated and stimulated to develop by parthenogenesis up to the blastocyst stage, indicating they are competent to transition toward the development of an embryo.

4. The author should give a deeper discussion about the limitation for this research.

This study had two limitations. First, there were only three treatments of TIMP1: 50, 100 and 150 ng/mL. The most popular function of TIMP1 protein is that it acts as an MMP inhibitor, which maintains the ECM necessary for COC expansion. Our results showed that COCs expanded structures were not favored by TIMP1 action. On the contrary, TIMP1 favored COCs compacted assembly. Figure 2B shows that the percentage of COCs expanded slightly increased in a TIMP1 dose-dependent (4.5 ± 5.9, 9.3 ± 4.6, and 16.3 ± 8.7 for 50, 100, and 150 ng/mL of TIMP1, respectively). This result may suggest that the TIMP1 concentration was insufficient to conduce to the COCs expansion. Hence, another studio is necessary using TIMP1 concentration higher than 150 ng/mL. Second, early embryo development was not measured after an in vitro fertilization (IVF) with a sperm. Although the parthenogenetic assay is an excellent way to evaluate the oocyte quality, an IVF is needed to evaluate the influence of TIMP1 over the yield of blastocysts.

Minors:

1.Line 63, the “in vitro” should be shown as italics, the author should check this for the whole manuscript.

Done

2. For figure 2A, 3A and 4A, the scale bar should be added.

Done

Reviewer 2 Report

This paper looks at the effects of adding human TIMP1 into in vitro maturation media to assess if it improved oocyte developmental competency. While certain markers are definitely improved, embryo development is only assessed via parthenogenesis and not actual fertilization. It is also not clear if the treatment is only relevant for oocytes and not embryos. However, the main claim is somewhat supported by the data.

The main confusing part is that in the discussion the authors keep referring to expanded and compacted COC. TIMP1 cannot do both. Those words are literal opposites of each other. So I don't know what they mean when the figure panels really show expansions and not compaction. 2B shows that 150 ng/ml has a slight increase in expansion but not much. There isn't that much difference in any of the bars for any of the treatments. Since the p-values are not on the graph this is hard to evaluate. Would be much better as a stacked bar graph (see below) so that it can be convincing. 

1. It is unclear whether the authors think that 50 or 150 ng/ml is better? Depending on the marker used, COC expansion, nuclear maturation etc, the results are different and it is hard to understand which is better. 50 ng/ml does give the largest amount of blastocyst so that may be preferable.

2. The representation of the graphs should be as a stacked bar graph with 4 bars for each treatment of drug, and each bar having stacks for the different phenotypes being scored. As of right now the graphs are extremely hard to understand and compare to each other. The significance values should be on the graph as asterisk as is usual. 

3. English needs a lot more editing and proofing. For eg. lines 167-171 do not make perfect sense so it is hard to understand the conclusion. 

4. Line 171: what is "unexpected" about the result?

Please take care to proof the manuscript (Line 171 has two control words. ). 

Author Response

RESPONSE TO REVIEWER 2 COMMENTS

This paper looks at the effects of adding human TIMP1 into in vitro maturation media to assess if it improved oocyte developmental competency. While certain markers are definitely improved, embryo development is only assessed via parthenogenesis and not actual fertilization. It is also not clear if the treatment is only relevant for oocytes and not embryos. However, the main claim is somewhat supported by the data.

We greatly appreciate the Reviewer's time to review our work and deeply value their opinions.

The parthenogenetic is a process where an oocyte, without sperm fertilization, can develop an embryo. This assay is excellent for evaluating the oocyte's capacity to use its cellular components to become an embryo.

We evaluated the influence of TIMP1 as a growth factor during the culture of COCs on the in vitro maturation media to improve oocyte development competence. A competent oocyte is an oocyte with quality because it is in the MII stage and has all the cytoplasmic components necessary for the transition to an embryo. After fertilization, the onset of embryonic development depends on the quality that the oocyte attained during competence development. The parthenogenetic is a process where an oocyte, without sperm fertilization, can develop an embryo. Therefore, a parthenogenetic assay is an excellent way to evaluate oocyte competence because there is no influence of sperm. With a parthenogenetic assay, all the cellular changes in the oocyte-to-embryo transition can be awarded exclusively to the oocyte.   Based on that, our results are relevant for the oocyte quality, not for embryos that develop after a sperm fecundation process.

The main confusing part is that in the discussion the authors keep referring to expanded and compacted COC. TIMP1 cannot do both. Those words are literal opposites of each other. So I don't know what they mean when the figure panels really show expansions and not compaction. 2B shows that 150 ng/ml has a slight increase in expansion but not much. There isn't that much difference in any of the bars for any of the treatments. Since the p-values are not on the graph, this is hard to evaluate. Would be much better as a stacked bar graph (see below) so that it can be convincing

We are thankful for the Reviewer's comment and respect their point of view about the discussion part over the dual effect (compaction and expansion) of TIMP1. For this reason, we reanalyzed these data and showed them as a stacked bar graph. After line 257, the Reviewer can see the new graph added to the manuscript. The Reviewer was right; the expanded COC structure slightly increased in 150 ng/mL treatment. A difference from culture control, where there was no statistical significance between compaction structures (P > 0.05); in all TIMP1 treatments, the COCs compacted were the most abundant. In the 50 ng/mL TIMP1 condition, the percentage of COCs compacted and partially compacted were close, while the COC expanded was the lower structure (57 ± 9.8% and 38 ± 8.6% vs. 4.5 ± 1.9% respectively; P < 0.05). Condition with 100 ng/mL of TIMP1 showed that the higher structure was COCs compacted compared with the COCs partially compacted and expanded assembly (64 ± 5% vs. 27 ± 6% and 9 ± 2% respectively; P < 0.05). The same was for the 150 ng/mL TIMP1 group; the COCs structure was more abundant than the others (50 ± 9% vs. 34 ± 8% and 16 ± 8%, respectively; P < 0.05). This paragraph is written in lines 207-216.

In the discussion section, we deleted the confusing part referring to the expanded and compacted effect of TIMP1. The discussion was focused on explaining the impact of TIMP1 on the compaction of the COCs during the IVM. The Reviewer can read this in lines 436-475.

1. It is unclear whether the authors think that 50 or 150 ng/ml is better? Depending on the marker used, COC expansion, nuclear maturation etc, the results are different, and it is hard to understand which is better. 50 ng/ml does give the largest amount of blastocyst so that may be preferable.

We agree with the Reviewer's opinion about the best TIMP1 treatment. To clarify and get an overview of how the observations of COC compaction structures, oocyte nuclear maturation, cortical pattern type, and parthenogenetic embryos are related to treatment variables, a Principal Component Analysis (PCA) was used to analyze multiple associations between them.

A Principal Component Analysis (PCA) was applied using a varimax rotation to reduce the dimensionality and detect essential patterns in data. Before analysis, the data were standardized, resting the media and divided by the standard deviation. After rotation with varimax, the sample adequacy was questioned by Kaiser-Meyer-Olkin (KMO) test (< 0.5), and Bartlett's test of Sphericity evaluated suitability for component analysis. The components were identified using an eigenvalue threshold of > 0.1. The result was shown in a biplot graphic (figure 6). The Bartlett test of Sphericity was significant (X2 = 35.6, df = 5, P < 0.001), which implies that the observations of COCs compaction structures, oocyte nuclear maturation, cortical pattern type, and parthenogenetic embryos were related to treatment variables and that PCA can be used to analyze multiple associations between them. The KMO test showed values between 0.7 and 0.8, indicating sample adequacy. The biplot graph displayed two principal components (PCs); PC1 absorbed 87% of the variables, while PC2 absorbed 7.4%; both add up to 94.4%; this implies sufficient information existed to interpret the data's most important aspects. In the biplot graph, the observations of COC expanded,  8-16 cells, 2-4 cells, and blastocyst embryos were located closer to each other with lower PC scores and were not associated with any treatment. Oocyte GV and oocyte GVBD had high PC scores but were not associated with any treatment. The observations for COC partially compacted and cortical granules pattern II were located together and were associated with the control culture (0 ng/mL) and 100 ng/mL TIMP1 treatment. The cortical granules pattern I and COC compacted were in the same place and were related to 50 ng/mL TIMP1 treatment. With these results, we inferred that the treatment of 50 ng/mL is the best condition to develop oocyte competence because its treatment promotes the formation of COCs compaction structure and the correct location of the cortical granules. Accordingly, we recommended supplementing the culture IVM media with 50 ng/mL

2. The representation of the graphs should be as a stacked bar graph with 4 bars for each treatment of drug, and each bar having stacks for the different phenotypes being scored. As of right now the graphs are extremely hard to understand and compare to each other. The significance values should be on the graph as asterisk as is usual. 

All the graphs were changed to a stacked bar graph.

3. English needs a lot more editing and proofing. For eg. lines 167-171 do not make perfect sense so it is hard to understand the conclusion. 

English editing and proofing were done.

4. Line 171: what is "unexpected" about the result?

The "unexpected" word was deleted.

Round 2

Reviewer 1 Report

For Major point 4, please add the discussion about the limitation in the revised manuscript. Besides this point, all the concerns that I raised have been addressed properly. 

Author Response

RESPONSE TO REVIEWER 1 COMMENTS

For major point 4, please add the discussion about the limitation in the revised manuscript.

We added the limitation of the study in the manuscript in lines 523-536

Besides this point, all the concerns that I raised have been addressed properly.

We are very grateful for the reviewer´s comments and greatly appreciate all the time spent reviewing this manuscript.